# Trauma clinicians' views of physical exercise as part of PTSD and complex PTSD treatment: A qualitative study

**Natasza Biernacka**, **Shivangi Talwar**, **Jo Billings***

Division of Psychiatry, University College London, London, United Kingdom

* j.billings@ucl.ac.uk

## Abstract

Physical exercise has the potential to be a helpful, adjunctive intervention for supporting people with post-traumatic stress disorder (PTSD) and complex PTSD (CPTSD). However, little is known about the views of clinicians on including physical exercise in treatment. We aimed to explore trauma clinicians' perspectives on the role of physical exercise in PTSD and CPTSD treatment and to understand key barriers and facilitators in recommending physical exercise as an adjunctive treatment. Twelve specialist trauma clinicians from across the UK were interviewed to explore their views on the role of physical exercise and the key barriers and facilitators in recommending it as an adjunctive treatment for PTSD and CPTSD. We used a qualitative explorative methodology with semi-structured interviews and analysed transcripts using reflexive thematic analysis. Trauma clinicians viewed physical exercise as a potentially beneficial supportive intervention for PTSD and CPTSD, and perceived several ways in which physical exercise could be included in the treatment process, with an individualised approach to care underpinning inclusion. However, there were also notable barriers to including exercise at environmental, client and clinician/service levels, including; limited access to exercise resources, client-related factors such as agoraphobia and physical health problems, clinician-related factors such as lack of confidence and training, and service-level factors such as gaps in service provision. The findings highlight the importance of considering individualised approaches to physical exercise interventions in trauma treatment. Addressing identified barriers, such as improving access to resources and providing training for clinicians, is crucial for successful integration of physical exercise into PTSD and CPTSD treatment protocols. This study underscores the need for further research to inform future policies and provide guidance for trauma clinicians on how to effectively incorporate physical exercise as an adjunctive treatment.

**Data Availability Statement:** Qualitative data from this study is not publicly available in line with the ethical approval for this study. Data can be made available upon reasonable request to the authors

## 1. Introduction

Post-traumatic stress disorder (PTSD) is defined by a cluster of physical and psychological symptoms which can arise as a result of exposure to a traumatic experience [1–3]. Diagnostic

via the UCL Research Ethics Committee
(ethics@ucl.ac.uk)

**Funding:** The authors received no specific funding
for this work.

**Competing interests:** The authors have declared
that no competing interests exist.

criteria for PTSD include persistent re-experiencing, causing a person to relive the traumatic event vividly and involuntarily; avoidance, such as of specific people or places which serve as reminders of the traumatic event; and hyper-arousal, commonly experienced as hypervigilance to current perceived threat [3]. Further difficulties may include adverse changes in cognition, mood and reactivity [1–3].

Complex post-traumatic stress disorder (CPTSD) is a recently recognised diagnosis in the International Classification of Diseases-11 (ICD-11) characterised by symptoms of PTSD plus an additional triad of symptoms: affect dysregulation, negative self-concept and interpersonal disturbances [3–5]. These include feelings of worthlessness, shame and guilt, difficulty regulating emotions, and problems connecting with others [3]. The experience of trauma in cases of CPTSD is more likely to have been either prolonged or repeated. It is often associated with childhood abuse or neglect, domestic violence, sexual abuse, war, slavery or torture [5–7].

Physical health symptoms and conditions commonly co-occur in PTSD and CPTSD [8] including poorer physical health-related quality of life, cardiovascular diseases, chronic lung diseases, gastrointestinal issues, greater musculoskeletal pain [9–11] sleep disturbances [12, 13], chronic pain [14] chronic fatigue [15, 16], autoimmune diseases [8], headaches [17], fibromyalgia, dizziness, chest pains [16], and cancer [16, 18]. Adverse childhood experiences (ACEs) have been found to be associated with negative physical health outcomes in adulthood, with patients who experienced four or more ACEs at significantly higher risk of diseases than people who had not been through an ACE [19]. Physical injury resulting from traumatic experiences can also be associated with PTSD [20, 21], with more severe injuries being associated with increased PTSD symptomatology [22].

Further physical effects of trauma on the body have been cited. Considering the neurobiology of PTSD, the presence and severity of PTSD symptoms have been associated with structural changes in the brain, including reduced volume in the hippocampus and anterior cingulate [23, 24] and reduced volume or thickness in the precuneus, insula, and prefrontal cortex [25]. These structural changes have been correlated with alterations of brain function, such as activation and connectivity in the amygdala and hippocampus [26–29] and functioning of the Hypothalamic Pituitary Axis (HPA), [30, 31]. A review suggests that the HPA response within someone diagnosed with PTSD is characterized by an exaggerated cortisol response [31] and increased inflammatory activity [32–35].

Modern medicine has tended to perpetuate a separation between body and mind, with advances in treatment for mental health difficulties focusing on developing pharmacological solutions and problem-focused talking therapies. Interventions for conditions directly caused by experiencing traumatic events, including PTSD and CPTSD, are no exception. However, people who suffer from the effects of trauma are unlikely to do so solely in their minds [36]. Symptoms, in most cases, manifest in clients' physiology or, in other words, "the body keeps the score" (ibid) of the experienced traumatic events. Indeed, in the past decade, there has been a growing interest in the effects of physical exercise on PTSD symptoms and the potential benefit of including body-focused therapeutic interventions in PTSD and CPTSD treatment.

Physical exercise refers to physical activity that is planned, structured, repetitive, and intended to improve or maintain physical fitness [37]. The WHO states that physical activity has significant health benefits, contributing to preventing and managing diseases such as cardiovascular diseases, cancer and diabetes and reducing symptoms of depression and anxiety [37]. Some research suggests that effectiveness is competitive to pharmacology and psychotherapy in improving overall cognitive function and well-being [38].

There is a growing evidence base for physical exercise as a supportive intervention for PTSD and CPTSD symptoms. In a narrative review of 19 studies that examined aerobic exercise and PTSD symptomatology, Hegberg et al. [39] reviewed nine observational studies and

ten intervention studies and concluded that physical interventions alone or as an adjunct to standard treatment might positively impact PTSD symptoms. In a meta-analysis of eleven studies across three countries, Björkman & Ekblom [40] concluded that physical exercise has the potential to be a helpful, supportive intervention, reporting a small to medium effect of exercise on PTSD symptom severity. Bjorkman & Ekblom [40] also reported positive effects of physical exercise on depressive symptoms, sleep disturbances, and substance use disorder. More recently, Jadhakhan et al. [41] conducted a systematic review of 13 studies from four countries and reported that combined exercise interventions had the best evidence for a beneficial effect on PTSD symptoms. None of the studies included specifically considered the impact of physical exercise on CPTSD symptoms.

There is some debate about what type, dosage and duration of exercise might be most helpful. Björkman & Ekblom [40] suggested that more significant amounts of exercise might provide more benefits. In their meta-analysis they explored whether there is a difference in the effect on PTSD symptoms between high- and low-intensity activities (yoga versus other exercises) or between group and individual exercises, and found no significant differences. The authors concluded that there is no established optimal dose, duration or type of exercise. In contrast, Jadhakhan et al. [41] explored which forms of exercise or physical activity have the most significant effect on PTSD outcome scores. They found that combined exercises (resistance training, aerobics, strength, and yoga) administered over 12 weeks, three times a week for 30–60 minutes, showed more significant effects on PTSD symptoms than individual forms of exercise.

The evidence on physical exercise as a supportive intervention for PTSD and CPTSD from systematic reviews and meta-analyses is still limited, possibly due to high heterogeneity in randomized control trials (RCTs) investigating various outcomes and using different measures of PTSD and CPTSD. In addition, adequately powered RCTs are required to provide more definitive evidence of a causal relationship between physical exercise and reduction in PTSD and CPTSD symptoms. Nevertheless, emerging evidence suggests that physical exercise could potentially play a role in PTSD and CPTSD treatment and could be an affordable, acceptable and scalable intervention, which could be included in future treatment guidelines.

To date, however, little is known about trauma clinicians' views on including physical exercise in treatment. Specialist trauma clinicians play a key role in deciding, together with their clients, what approach to take to treatment and the phasing and sequencing of interventions [42]. Understanding specialist trauma clinicians' views about exercise, whether this is something they endorse, and why or why not, is therefore crucial in considering the potential inclusion of physical exercise interventions as an adjunct to treatment of PTSD and CPTSD. Thus, this paper had the following research aims:

1. To explore trauma clinicians' perspectives on the role of physical exercise in PTSD and CPTSD treatment.

2. To understand trauma clinicians' perceptions of the key barriers and facilitators to recommending physical exercise as a supportive treatment for PTSD and CPTSD.

## 2. Methodology

### 2.1. Ethics

Ethical approval for the study and its procedures was obtained from the Research Ethics Committee at University College London (Ref. 23469.001).

## 2.2. Design, participants and procedure

The study consisted of individual semi-structured interviews with specialist trauma clinicians. Participants were qualified mental health clinicians working in the UK, specifically treating clients diagnosed with PTSD and CPTSD in dedicated outpatient trauma services. In the UK, specialist trauma services are usually tertiary level services, and the clinicians working within them are experienced psychological therapists, who would be key in working with their clients to make decisions about treatment approaches, content and sequencing.

We used purposive and snowballing approaches to recruit participants who were all specialist trauma clinicians working in specialist tertiary trauma services across the UK. Potential participants were initially contacted by JB (Consultant Clinical Psychologist and Professor) through UK wide professional trauma networks (i.e. the UKPTS) or via social media platforms (Twitter, LinkedIn, and Facebook). Clinical contacts of JB were also asked to circulate details of the study to other clinical colleagues in specialist trauma services. Clinicians interested in participating in the study were followed up by NB (lead researcher), who forwarded the participant information sheet, consent form, and sociodemographic form. All participants provided written informed consent prior to taking part in the study. Participants also provided sociodemographic information regarding gender, age, ethnic background, occupation, current work setting, and region of the UK. Interviews were conducted remotely online by NB. Recordings were transcribed by NB and any identifying features of the participants, such as their roles and organisations, clients, or colleagues, were removed. Pseudonyms are used in the presentation of the results.

## 2.3. Interview guide

The interview guide was developed collaboratively by the research team and based on the study's research questions. The semi-structured interview consisted of an initial question about the participant's clinical work followed by their views about recommending physical exercise as a supportive intervention for PTSD/CPTSD, and what barriers and facilitators they perceived to delivering such interventions (see S1 Text).

## 2.4. Data analysis

Reflexive thematic analysis was used to analyse the data [43, 44]. As the nature of this study was exploratory, this approach allowed us to capture a variety of opinions and ideas which could help inform future clinical practice and further research. The coding of the interview data was semi-inductive and semi-deductive and was led by NB. ST independently coded 10% of the transcripts to help redress any blind spots NB might have about this topic. The codes were frequently discussed among the research team. This helped to generate other novel codes, which were subsequently incorporated into the coding framework and final themes.

## 2.5. Reflexivity

As a qualified yoga teacher NB came to the research project with a pre-established belief that physical exercise is beneficial. NB was transparent about her prior assumptions and those were addressed during research team discussions which tempered NB's pre-conceived ideas, encouraged her to retain curiosity, and to deliberately look for exceptions in the data. ST is a Clinical Psychologist and academic researcher with experience of working clinically with people with PTSD and CPTSD. JB is a Consultant Clinical Psychologist and Professor with over 20 years of experience of working clinically with PTSD and CPTSD. As a research team, we brought diversity to the research data from different ethnic backgrounds and career stages. We

have approached this topic from a critical realist stance, fitting with the underpinnings of reflexive thematic analysis.

## 2.6. Quality

We have endeavored to ensure the highest quality of reporting in this qualitative study, adhering to the Standards for Reporting Qualitative Research (SRQR) [45] a list of 21 items considered essential for complete and transparent reporting of qualitative research. We have further ensured the trustworthiness and transparency of our research by discussing evolving themes with clinical academic peers, and sharing our preliminary findings with participants as a means of validity checking.

## 3. Results

Twelve UK based specialist trauma clinicians participated in this study. The gender, ethnicity and roles of participants are shown in Table 1.

Interviews took place between August and October, 2022 and lasted between 14 and 27 minutes.

We identified three main themes relating to clinicians' views on the inclusion of physical exercise in PTSD and CPTSD treatment. Using a semi-inductive, semi-deductive approach,

**Table 1. Participant characteristics.**

| Participant Characteristics | Number of participants (%) |
|---|---|
| **Gender** | |
| Female | 9 (75.0%) |
| Male | 3 (25.0%) |
| **Age** | |
| <30 | 2 (16.7%) |
| 30–39 | 3 (25.0%) |
| 40–49 | 3 (25.0%) |
| 50–59 | 3 (25.0%) |
| 60+ | 1 (8.3%) |
| **Ethnicity** | |
| White | 12 (100.0%) |
| **Occupation** | |
| Clinical Psychologist | 9 (75.0%) |
| Counselling Psychologist | 1 (8.3%) |
| Counsellor / Psychotherapist | 1 (8.3%) |
| CBT Therapist | 1 (8.3%) |
| **Setting worked in** | |
| National Health Service (NHS) | 6 (50.0%) |
| Private Practice | 3 (25.0%) |
| University | 2 (16.7%) |
| NHS and Private Practice | 1 (8.3%) |
| **UK geographical region** | |
| London | 5 (41.7%) |
| South East | 2 (16.7%) |
| South Central | 2 (16.7%) |
| South West | 2 (16.7%) |
| National | 1 (8.3%) |

**Fig 1. Themes and subthemes explaining clinicians' view on physical exercise in PTSD/CPTSD treatment.**

the themes identified included potential benefits of physical exercise, barriers to including physical exercise, and individualised care underpinning the benefits and barriers. The relationships between themes are illustrated in Fig 1.

## 3.1. The potential benefits of physical exercise

All participants saw value in physical exercise. However, there were variations in their perception of its importance and implementation in treatment.

**3.1.1. Bare necessities: an integral part of treatment.**   Some trauma clinicians viewed physical exercise as part of generic health advice, not specific to survivors of traumatic events.

*"Physical exercise is wonderful for any mental health issue. It is really a basic part of self-care."*

- Lucy, Clinical Psychologist

Other participants were more deliberate about recommending physical exercise to trauma-affected clients and saw it as an integral part of the treatment process, helping clients to recover better and quicker, tackling the effects of trauma on the physical body and the mind. For some clinicians, like Emma, exercise was routinely discussed with clients throughout therapy as they saw exercise as an essential part of the recovery process.

*"(. . .) all of those things are under regular review because we know they get better quicker if they do those things, and that is I guess . . . we measure that. So, you know, we can say with, you know, some certainty in the evidence that exercise is helpful."*

- Emma, Psychologist

**3.1.2. Mind and body: Holistic treatment.** Many participants perceived holistically attending to client's physical and psychological wellbeing as essential and inseparable. They described bringing the topic of trauma and the body into psycho-education sessions with the clients.

*"It is always about doing the two. (. . .) So, you work with the psychology of it, but you also work with the physiology of it. Actually, together, it shows that that person is safe, that nothing is happening to them."*

- Emma, Psychologist

Clinicians described the importance of intentionally introducing exercise so that it is not "exercise for the sake of exercise" but rather a form of mindful action. Some of them explained that they would follow up with clients on the activities engaged in either outside or during their sessions. They would explore with clients about their feelings, sensations and experiences while exercising.

*". . .for me, it is very much about connecting with your body and your physiological experience, as well. So it is not just about the activity. It is also about noticing what you're feeling, where you're feeling it in your body, what that experience is like."*

- Peter, Psychotherapist

**3.1.3. Reclaiming life and the body.** Physical exercise was perceived as a tool for clients reclaiming goals around their life and their bodies. Many forms of physical exercise were used in treatment as a "vehicle" to help clients reclaim or rebuild their lives, including yoga, running, boxing, walking, resistance training, body weight exercises. Where physical exercise was not necessarily the goal in itself, it played an integral part in facilitating goals such as socializing or stepping outside their homes.

*"So there is the physical exercise part but is it the intrinsic physical exercise that's the important bit, or is it all the stuff that comes around it, like the social structures or the things like people doing park run."*

- Josh, Clinical Psychologist

Some clinicians were of the view that exercise enabled their clients to tolerate this reconnection with their bodies. Further, some clinicians found exercise an effective way to help clients feel grounded and safe. Participants also described how some clients hold onto the trauma in their bodies, and how working on physical awareness with their clients helped release that physical trauma.

*"That feeling of body boundaries and that sense of I don't know, I guess sort of strength, and not being, you know, not being, intruded upon."*

- Ellen, Clinical Psychologist

**3.1.4. Reducing arousal.**   Clinicians found it helpful to use physical exercise to enable their clients to reduce symptoms of hyper-arousal and hyper-vigilance, and to manage anxiety.

*"I also felt it would help him with that constant shaky sense of being vigilant and on guard. So, it would help to make that anxiety lower. And I will say, I also think of a similar client, (. . .) he started by doing it, using his treadmill or at home, and for both (. . .) of those clients, it was definitely helpful."*

- Natalie, Clinical Psychologist

**3.1.5. Aiding bilateral processing.**   A few clinicians mentioned purposefully using physical exercise for its possible benefit in processing trauma memories. They thought that some forms of physical exercise that engage both sides of the body may be more advantageous as they can mirror the processes of EMDR.

*"It also fits alongside for me something of EMDR, as well. So, moving or running, it's about bilateral stimulation, so you are activating the left and right-hand sides. (. . .) so walking is good for that kind of movement that activates bilateral stimulation."*

- Peter, Psychotherapist

## 3.2. Barriers to including physical exercise

Participants described barriers to introducing and recommending physical exercise as a part of their clients' treatment. We divided these barriers into environmental, client, clinician and service-related roadblocks.

**3.2.1. Environmental barriers.**   *Access.* Clinicians expressed a need for increased access to exercise. All participants stated that socio-economic factors such as access to funds, and income status, both on the client and service side, play a crucial role. While working with vulnerable clients, clinicians described that clients often do not have access to resources such as running shoes or gym passes. A few participants mentioned an "exercise on prescription" initiative that allowed General Practitioners to prescribe exercise and enabled clients to have discounted access to gyms. However, participants were not aware whether the program was ongoing. Clinicians from uni-disciplinary psychology services said that they would need to check any facilities to refer their client for physical exercise and were unsure of the practicalities of such referrals.

*" [vulnerable clients] don't have those resources and feel isolated. They are the ones that need this. But we need funding. You need the funding, for the support workers, the kind of safe spaces in the gym, for training people in the gym or outdoors or, you know, wherever it is, having a range of different approaches, and different things for different ages."*

- Ellen, Clinical Psychologist

*Negotiating the environment*: *The need for trauma-informed and culturally appropriate exercise resources.* Clinicians explained how gyms and swimming pools could be challenging for clients to negotiate. Such environments are often loud and tend to be predominantly occupied with males, which could be distressing to abuse survivors. Trauma-informed exercise spaces where clients could receive appropriate support from trained staff, if they potentially became triggered, were suggested as a crucial resource to support clients' needs. Accessing trauma-

informed spaces and forms of physical exercise, such as yoga or classes adapted to the client's specific needs, were perceived as vital for clients to benefit from physical exercise.

*"There is something quite challenging about being, say, in a swimming pool or a gym environment, you know, there is a lot, for women, there is a lot of men around, your body is quite on display, and I think that for people who have been abused, of it there is like permanent scarring, things like that, I think that can be quite a challenge."*

- Lucy, Clinical Psychologist

Some clinicians reflected on the role of clients' cultural backgrounds in engaging in exercise, and the need to take this into consideration. The need for trauma-informed resources interacted with the need for the availability of culturally-informed resources. For example, as explained by one trauma clinician, a participant might need an exercise to be adapted for them due to physical health constraints. In addition, considering their cultural background, they may face language barriers and may be unable to explain to the trainer that they need the exercise to be adapted. Therefore, culturally-tailored practices would be a general requirement when working with this client group and an important step towards exercise interventions being trauma-informed.

*"(. . .) if they spoke English, they could probably do something that someone could support them with by explaining adaptations but because they don't, you know, how is . . . how are they going to go to their local yoga class and have someone explain how to adapt it for them?"*

- Mary, Clinical Psychologist

**3.2.2. Client factors as barriers to including physical exercise.** Trauma clinicians identified client factors that created barriers to engaging with physical exercise. Those factors interacted with the environment.

*Agoraphobia.* Most clinicians described working with clients who find leaving the house challenging and anxiety-provoking. In those cases, online and home workouts were mentioned as an option, granting the clients access to privacy in which to exercise. Although environmental factors, such as space also interfered here.

". . .many my clients avoid leaving the house completely, just because they are so anxious about being triggered by things like noises, airplanes, certain smells, (. . .) if you think of exercise, you might be thinking to leave the house. Obviously, you can do work at home, but if your house is quite small, you're quite limited, aren't you."

- Vicky, Clinical Psychologist

*Triggers within exercise.* Some of the clinicians described how exercise could be triggering for many of their clients. Clinicians explained how natural physical changes that occur during exercise could be associated with the traumatic experience, such as labored breathing. Of note, participants explained seeing an opportunity in physical exercise to work with those triggers in therapy and desensitize clients to these cues.

*"You don't even need to go out to do exercise, but it's just being aware, really, of the triggers that might be in it (. . .). And sometimes if they are doing exercise that there will be something triggered. But then you deal with that in the next session to find out, well, why did it happen?*

*(. . .) Once you know what it is, then you can desensitise them from that thing whatever it was."*

- Emma, Psychologist

*Physical health problems.* Clinicians, especially those working with CPTSD, described physical health symptoms as a barrier that could stop clients from engaging in exercise. Clinicians recognised that physical exercise could be supportive in improving their physical health, but within managed expectations of potential achievements specific to a client.

*"I think the big thing would be pain and physical ability. I mean obviously you can anything (. . .) even if we're going for a walk, a short walk is better than nothing."*

- Vicky, Clinical Psychologist

*Depression.* Clinicians mentioned that clients often experience comorbid depression, which obstructs their motivation to engage in daily activities.

*"They just do not have . . . if you are really, really low and suffer depression, you just don't have the motivation to get up, and you certainly don't have motivation beyond kind of what is minimally required. "*

- Vicky, Clinical Psychologist

Clinicians recognized that introducing exercise may be exceptionally challenging when the client does not feel motivated or is not able to exercise. However, they reflected this may be a "chicken and egg" situation where those problems could be alleviated to some extent with the help of exercise. In the case of physical health, clinicians felt that they lacked the knowledge to know when it is safe to push the client and how much encouragement would be safe. Further, participants recognized how recommending something the client feels they cannot do might be disheartening for the client, adding to their experience of shame.

*Not seeing the value in physical exercise.* Clinicians described that many clients struggled to see the value in physical exercise. As previously mentioned, they may either lack motivation or may not be able to function at their premorbid level.

*"People are quite demotivated because, you know, if you are asking them to go for a walk, say, and they used to, you know, do something a lot more athletic, I think that is kind of "what is the point", "cannot be bothered."*

- Lucy, Clinical Psychologist

In contrast, one clinical psychologist mentioned that clients who do not have a history of being physically active may not have previously experienced exercise being beneficial and may struggle to see the value in it for that reason.

*"If they do not necessarily have physical exercises as a historical pattern, and I think it's harder for them to be motivated or to understand why it might help because they haven't got that experience."*

- Natalie, Clinical Psychologist

Some trauma clinicians mentioned that clients may not see the value in physical exercise as it may not historically have been a priority, compared to their struggles to meet more basic human needs such as food and shelter, as was the case in many trauma survivors' lives.

*"I guess you were living in very kind of unsafe, unstable parts of the world. I imagine if you're focused on you know, food, shelter, and keeping your family safe, exercise is probably not high up your priority list, might not be anything you even necessarily prioritised in your life."*

- Lucy, Clinical Psychologist

*Shame*. As one participant described, clients with CPTSD often experience shame and consequently would not see self-care as essential for survival. Feelings of shame interacted with the barrier of the environment which further underlines the need for trauma informed and culturally adapted exercise spaces.

*"I think a lot of our clients who have got shame and complex PTSD, obviously their wellbeing and physical wellbeing is not a priority (. . .). It's that self-care, wellbeing, physical health is just not one of their key values,(. . .) they haven't grown up to be shown or taught that taking care of yourself is really key."*

- Barbara, CBT Therapist

**3.2.3. Clinician and service-level barriers to including physical exercise.** *Exercise being missed*. Some clinicians reflected that these research interviews were the first time someone had inquired about their perspective on exercise as part of PTSD/CPTSD treatment.

*"I think you know; this is the first time that I've been asked in terms of physical exercise, I think it just needs to somehow sort of be brought into therapists awareness, a little bit more."*

- Barbara, CBT Therapist

Some participants acknowledged that they had not previously had opportunity to think about physical exercise as part of treatment. Throughout the interviews, we noticed that a few participants seemed quite ambivalent about the topic of physical exercise at the beginning of the interviews, however, towards the end, came up with solutions and feasible options for their service to implement and engage clients with physical interventions. One participant reflected that they had never thought about exercise as an intervention in itself, nor previously explored the benefits that exercise could bring. As a result, they decided to bring this up in their supervision.

*"It is something worth me thinking about taking to my own supervision (. . .). I do like approve and encourage clients to think of like physical activity as part of their reclaims. But I don't yeah, like, I said, I don't think of it as like physical activity as an intervention, which maybe is different."*

- Mary, Clinical Psychologist

Clinicians who deliberately included physical exercise in treatment expressed frustration around exercise being a "frequently missed" area. Clinicians also reflected on having limited time to spend with clients which, in turn, impacted their ability to implement physical exercise interventions. For example, one participant described feeling pressure to start trauma

processing work as soon as possible and rarely finding time to address physical exercise. In contrast, another clinician mentioned that limited contact with clients is precisely why she encouraged her clients to exercise, believing this helped them to recover better and quicker.

*"I'm really surprised by this (. . .) I think, you know, therapists don't kind of prioritise it enough at all, so that might be a block in terms of the therapists trying to promote it. (. . .). I think it's really missed. And we're very pharmaceutical, one hour and a therapy room kind of based, and I think it saddens me that clients would get so would get better quickly if they had access to physical exercise."*

- Barbara, CBT Therapist

Some participants were more cautious than others to recommend physical exercise because they thought that their training did not equip them with the skills to recommend it, often due to their clients' comorbid physical health problems.

*"I think sometimes it can feel a bit deskilled in terms of knowing how much to push. In line with that, you don't want to exacerbate a physical condition, but at the same time, we know that a lot of things like chronic pain, fibromyalgia, any . . . like some kind of pace activity is actually more beneficial, but I guess it's not necessarily having the knowledge to know how much to push that."*

- Lucy, Clinical Psychologist

*Gaps in service*. Clinicians working in uni-disciplinary psychology services described lack of access to medical advice as a limiting factor in recommending exercise, as well as no access to facilities where they could refer clients or to adequately trained or designated staff. Some participants explained they would find a multidisciplinary team approach helpful in addressing physical exercise and the client's physical and mental health, including personal trainers and occupational therapists.

*"I could kind of see that working similarly in a multidisciplinary team that somebody might be having trauma-focused CBT but also be seeing a sort of physical health, this can be an interventions worker alongside. As long as you're kind of having, you know, this multidisciplinary discussion to ensure that your work is complementing one another (. . .) then yeah, I think that could work really well, in terms of incorporating it into trauma-focused CBT."*

- Veronica, Clinical Psychologist

*Lack of evidence and policies*. Clinicians mentioned that having access to more evidence, policies and guidelines regarding exercise would enable them to feel more confident as to when it is appropriate to recommend physical exercise in treatment. More easily accessible evidence could also motivate clients to engage in physical exercise more. For example, clinicians commented that having easy-read pamphlets for clients could be helpful.

*"I think, having some kind of service policy, or even like an NHS generic policy for these the kind of exercises that are recommended. These are criteria, for when it wouldn't be appropriate if in doubt, this is what to do would be really helpful."*

Mary, Clinical Psychologist

## 4. Discussion

In this study we explored specialist trauma clinicians' views regarding the adjunctive use of physical exercise in PTSD and CPTSD treatment and their perceptions of potential barriers and facilitators to including physical exercise as a supportive intervention. Potential benefits of physical exercise were perceived in holistically addressing clients' health issues and treating the effects of trauma through physiological and psychological pathways. Some clinicians deliberately recommended exercise to treat physiological symptoms such as hyper-arousal. Some viewed it as general health advice that is part of maintaining wellbeing. Clinicians often brought up including physical exercise in reclaiming life goals, using it as a vehicle for achieving goals via socialising or getting outside more, rather than necessarily exercise being the goal itself.

Emerging evidence suggests that physical exercise can modulate neurobiological mechanisms implicated in stress responses, such as reducing cortisol levels and enhancing neuroplasticity [46]. Indeed, a recent review found that physical activity showed a good protective effect in those who engaged in high level of physical activity before traumatic events [47]. Integrating physical exercise into trauma-focused interventions may thus capitalise on its potential to mitigate the neurobiological alterations associated with trauma exposure, offering a multifaceted approach to addressing PTSD and CPTSD.

The findings of our study are consistent with those of Björkman & Ekblom [40] who concluded that exercise has the potential to be a helpful, supportive intervention for treating PTSD, with the clinicians we spoke to reporting physical exercise as beneficial for their clients in many ways. However, there were varied ideas amongst trauma clinicians about the potential benefits of physical exercise and, therefore, incorporating it in clients' treatment. For example, whether it should be deliberately included to help alleviate specific symptoms and reconnect clients to their bodies, as general wellbeing advice, or as a reclaiming life goal. Perhaps, such breadth of views emerges from the participants' creativity and ability to meet the needs of a vastly heterogeneous group of clients.

Björkman & Ekblom [40] recommended that future research investigates the optimal dose and type of exercise and suggested that more significant amounts of exercise may provide more benefits. A clear and opposing finding of the current research study is that experienced trauma clinicians believed that physical exercise in any form should be individually tailored to each client. Therefore, looking for an optimal dose or type of exercise could be counterproductive in light of the findings of this study. Clinicians reported that clients seemed to respond differently to various forms of physical exercise, with some benefiting from boxing and some finding running or yoga useful. There might be some clients who find physical exercise redundant. The current study underlines that clinicians need to meet the clients where they are regarding the intensity of exercise and the nature of the physical activity, bearing in mind the specific trauma(s) they may have experienced.

The current study is in contrast to the findings of the systematic review by Jadhakhan et al. [41] that aimed to determine the optimal form of exercise that most significantly affects PTSD outcomes. Jadhakhan et al. [41] concluded that combined exercise interventions administered over 12 weeks, three times a week for 30–60 min, showed more significant effects on PTSD symptoms than individual (non-combined) forms of exercise. Given the variety of functions which physical exercise may serve in recovery from PTSD and CPTSD, as highlighted by the specialist trauma clinicians in this study, such recommendations may be unhelpfully reductionistic. Defining an optimal type, dose, duration and intensity of exercise may therefore not be possible nor desirable. In the views of our participants, the inclusion of physical exercise as an adjunctive intervention should be underpinned by the principle of an individualised approach to care.

There was a clear consensus amongst the trauma clinicians we spoke to that engaging their trauma-affected clients in physical exercise could benefit their wellbeing and help to alleviate symptoms of PTSD and CPTSD. Clinicians also perceived that there were environmental barriers interacting with several challenges that clients may face in the process of treatment. It is imperative to note that each setting would not be adequately prepared or resourced to facilitate helping trauma-affected individuals and every clinician may not feel skilled to recommend physical exercise to their clients or able to find the time to do so in time-limited therapy.

Some of the barriers identified in our study share similarities with those observed in other mental health disorders [48, 49]. For example, access to resources and facilities has been recognized as a common barrier to exercise participation among individuals with various mental health conditions [50]. Additionally, environmental factors, such as the atmosphere of exercise spaces, can act as triggers for individuals with PTSD, similarly contributing to exercise avoidance as observed in other disorders [51]. Lack of motivation and low mood are prevalent barriers across different mental health disorders, indicating a shared challenge in promoting physical activity within these populations [49, 52]. Other barriers specific to people with PTSD/CPTSD were also identified in our study. Triggers related to specific traumatic experiences may lead to re-experiencing symptoms and strong emotional and behavioural reactions, making it particularly hard for this client group to engage in certain types of physical activities [53]. Specific to trauma survivors, especially those with interpersonal trauma experiences, shame could further impact willingness to participate in exercise interventions, particularly those in shared exercise spaces [54]. These findings suggest that while there are shared barriers to exercise across mental health disorders, there are some unique PTSD-related barriers which highlight the need for tailored approaches in promoting physical activity within this population.

Our study found that many of the barriers interact with each other, which further contributes to their maintenance. Similar findings were reported in a recently published qualitative study exploring healthcare professionals' views of barriers to delivering trauma-focused interventions for people with psychosis and post-traumatic stress disorder [55]. Similar barriers included staff confidence, knowledge and beliefs. In the current study, some clinicians explained experiencing a lack of confidence and knowledge in recommending physical exercise, whether due to lack of access to sufficient evidence and policies that would provide guidance or lack of training in physical health interventions. Some clinicians were cautious of recommending exercise due to their beliefs about client barriers, such as physical health issues or possibly evoking feelings of shame. Other similar barriers related to structural support, notably service configuration. In this study we also identified gaps in services as a barrier, with professionals in uni-disciplinary psychology services describing lack of access to medical advice as a limiting factor in recommending exercise and a lack of clarity about the practicalities of making referrals for physical exercise. Another study focused on the implementation of intensive treatments for PTSD identified barriers similar to our current study [56]. Namely, staff attitudes, such as lack of their early "buy-in" to the effectiveness of the provided care, limited access to resources, and lack of flexibility within the system are in accordance with our findings, especially lack of confidence in clinicians, limited access to time and funds, and systemic limitations due to gaps in services.

Our study findings highlight the need for trauma-informed and culturally adapted exercise spaces. Cultural safety within trauma and violence informed care (TVIC) would ensure culturally sensitive and equitable opportunities to engage in exercise interventions (Browne et al., 2015). This entails more than just the physical layout of spaces such as gyms; it requires a comprehensive approach that considers cultural factors, language preferences, and potential triggers for individuals with PTSD and CPTSD [5, 53]. In such spaces, gym staff would receive specialised training in

recognising signs of distress and responding with empathy and sensitivity [53]. For example, staff could be trained to provide verbal reassurance, offer a safe space for individuals to take a break, or provide access to supportive resources if needed [53]. A potential barrier for sustaining staff competency would be access to additional funding. Nature-based activities, such as hiking or outdoor group workouts, can provide opportunities for grounding, relaxation, and connection with the natural environment, which may be particularly beneficial for trauma survivors [57]. Therefore, a trauma-informed approach to exercise interventions should consider both indoor and outdoor-based activities, adapting the environment and support mechanisms to meet the diverse needs of individuals while promoting physical and mental well-being.

## 4.1. Strengths and limitations

To our knowledge, this is the first study to explore clinicians' views regarding physical exercise as a supportive intervention in PTSD and CPTSD treatment. The literature on this topic is scarce, and this study provides some preliminary evidence to add to the growing body of research regarding the importance of addressing an individual, as a whole, in the treatment of the effects of trauma. We accessed a range of specialist trauma clinicians working across the UK. The analysis was conducted rigorously, and the validity of analyses was maximized, including sharing results with participants and a panel of peer trauma researchers. The research team was diverse, composed of researchers from different career stages and cultural backgrounds.

There are, nevertheless, important limitations to this study that need to be considered. This study presents the views of only 12 clinicians recruited through purposive and snowball sampling approaches relying on professional networks. Other clinicians may have had different experiences and views which we were not able to access via this approach. Further research is needed to determine whether these views are representative of the wider network of trauma professionals. Interviews were relatively brief, necessitated by how busy the trauma clinicians we spoke to were. More in-depth interviews would allow greater exploration of many of the issues identified. The study included clinicians practicing in the UK and was conducted by researchers working at a UK university. Despite the best efforts of the research team to engage a diverse population of participants at the recruitment stage, the sample of clinicians willing to participate was homogenous with all participants identifying as White. This study therefore presents a UK-based and Western-specific view of physical exercise as part of PTSD and CPTSD treatment. The transferability of our study's findings is limited by the training and experience of UK clinicians. Further research in other cultural contexts is needed. This field of research will also be enhanced by including service users' views on incorporating physical exercise as an adjunct to treatment for PTSD and CPTSD.

## 5. Conclusions

This study provides a preliminary analysis of specialist trauma clinicians' views of the role of physical exercise in treatment for PTSD and CPTSD. We found that there were general and specific perceived benefits and barriers of including physical exercises in trauma focused treatments. Clinicians recognized that each service user's needs were unique, and that the principle of individualized care should underpin delivery. Further research is needed to determine how physical exercise could better fit into treatment protocols, inform future policies and provide guidance for trauma clinicians.

## Supporting information

**S1 Text. Interview guide.**
(DOCX)

## Acknowledgments

We would like to thank the trauma clinicians who generously invested their time to take part in this study.

## Author Contributions

**Conceptualization:** Natasza Biernacka, Shivangi Talwar, Jo Billings.

**Data curation:** Natasza Biernacka.

**Formal analysis:** Natasza Biernacka, Shivangi Talwar, Jo Billings.

**Investigation:** Natasza Biernacka.

**Methodology:** Natasza Biernacka, Shivangi Talwar, Jo Billings.

**Project administration:** Natasza Biernacka.

**Supervision:** Shivangi Talwar, Jo Billings.

**Validation:** Natasza Biernacka, Shivangi Talwar, Jo Billings.

**Visualization:** Natasza Biernacka.

**Writing – original draft:** Natasza Biernacka.

**Writing – review & editing:** Shivangi Talwar, Jo Billings.

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
