## [Decision Letter · Decision Letter 0]

19 Mar 2024

PMEN-D-24-00076

Trauma clinicians' views of physical exercise as part of PTSD and CPTSD treatment A qualitative study

PLOS Mental Health

Dear Dr. Billings,

Thank you for submitting your manuscript to PLOS Mental Health. After careful consideration, we feel that it has merit but does not fully meet PLOS Mental Health’s publication criteria as it currently stands. Therefore, we invite you to submit a revised version of the manuscript that addresses the points raised during the review process.

We ask you to expand the abstract for clarity on identified barriers and benefits, provide comprehensive details on study design, methodology, and data analysis. It's essential to deepen the discussion around the role of trauma clinicians, compare barriers to those in other mental disorders, and describe a trauma-informed exercise environment in detail. Incorporating client feedback and discussing neurobiology's role in PTSD/CPTSD treatment will strengthen your manuscript. Please also correct any statistical and grammatical issues. Your revised submission should address these concerns in a consolidated manner for re-evaluation.

We look forward to receiving your revised manuscript.

Kind regards,

Hongru Zhu

Academic Editor

PLOS Mental Health

Journal Requirements:

1. Please update your online Competing Interests statement. If you have no competing interests to declare, please state: “The authors have declared that no competing interests exist.”

2. Please ensure that the Title in your manuscript and the Title in your online submission form are the same.

3. Please provide separate figure files in .tif or .eps format only and remove any figures embedded in your manuscript file. Please also ensure that all files are under our size limit of 10MB. You may leave the figure captions or legends in the manuscript.

4. We have noticed that you have uploaded Supporting Information files, but you have not included a list of legends. Please add a full list of legends for your Supporting Information files after the references list.

Additional Editor Comments (if provided):

Reviewers' comments:

Reviewer's Responses to Questions

**Comments to the Author**

1. Does this manuscript meet PLOS Mental Health’s publication criteria? Is the manuscript technically sound, and do the data support the conclusions? The manuscript must describe methodologically and ethically rigorous research with conclusions that are appropriately drawn based on the data presented.

Reviewer #1: Yes

Reviewer #2: Partly

Reviewer #3: Yes

2. Has the statistical analysis been performed appropriately and rigorously?

Reviewer #1: N/A

Reviewer #2: Yes

Reviewer #3: Yes

3. Have the authors made all data underlying the findings in their manuscript fully available (please refer to the Data Availability Statement at the start of the manuscript PDF file)?

Reviewer #1: Yes

Reviewer #2: Yes

Reviewer #3: Yes

4. Is the manuscript presented in an intelligible fashion and written in standard English?

Reviewer #1: Yes

Reviewer #2: Yes

Reviewer #3: Yes

5. Review Comments to the Author

Reviewer #1: The article is great, i enjoyed reading it :) The only thing is to expand a bit the abstract, in the Result section, so it will be more clear from the beginning, what kind of barriers were identified, and what kind of benefits.

Reviewer #2: Thank you for allowing me the opportunity to review this paper. I am however not sure that I can recommend this article for publication. I am not sure that the article would be of substantial interest or applied importance.

The article outlines the views of clinicians re the integration of exercise into the treatment of those individuals with PTSD or CPTSD. However, I have several thoughts.

Firstly, the sample is small. There is no mention of data saturation. Secondly, the article would benefit from more depth re the role of these trauma clinicians in determining treatment programs. In other words, are these the group of professionals who decide treatment approaches for the client.

The barriers and facilitators outlined are as would be expected. It would benefit from a discussion of how these compare and contrast with those relevant to clients with other mental disorders. In other words, are individuals with PTSD or CPTSD unique in any way?

There is talk of a trauma informed exercise environment? What would this look like? There is talk about individuals within the gym being able to support the individual if they were triggered. What would this look like? Would this require training to be put out across gym staff? Is this feasible? It would benefit from more of an in-depth discussion of gym exercise versus outdoor activity given the symptom profile associated with PTSD and CPTSD.

I know that there is a discussion of client factors, but it would also benefit from evidence from clients. Overall I just felt that this paper needed much more depth in order to be of applied importance. I wish the authors good luck in their future research

Reviewer #3: General comments and some key concerns:

1. It is an interesting study that is giving an insight on the Trauma clinicians' views of physical exercise as part of PTSD and CPTSD treatment. All along in literature, Exercise has been documents for be crucial in maintaining body and brain health with few studies done to support this. So any study like this one is vital in advancing the knowledge in this area. However, below are my comments.

2. Title

• Avoid abbreviations in the title – Write “PTSD and CPTSD” in full

3. Abstract

• What was the study design?

• How was the interviews conducted to collect the data?

• Conclusion is not based on the key findings of the study

3. Methods section

• How were the specialist traumas clinicians identified and recruited in the study? How was the study participants selected?

• In data collection, was the structured interviews conducted using a designed tool as a guide? Were responses to saturation able to be reached??

• The statement “Clinical contacts were also asked to circulate details of the study to other clinical colleagues in specialist trauma services----“What type of study design is this? The study design needs to be clearly explained!!!!!

• What were the variables captured?

• Which statistical package was used to summarize the data if any?

• How was the ethical issues handled i.e. study approval and number if possible or waiver was given?

4. Results

• Responses in table 1should be reported as % (n) or add another column for %.

• Characteristic item on “Occupation”, there was no Psychiatrist!!!

• 3.1. The Potential Benefits of Physical Exercise -All participants saw value in physical exercise. What were those values mentioned?

• 3.1.2. Mind and Body: holistic treatment – same as above

• 3.1.3. Reclaiming Life and the Body

• Physical exercise was perceived as a tool for reclaiming goals around life and their bodies. Many forms of physical exercise were used in treatment – what were those different forms encountered?

5. Discussion

• Discussion should also bring in the role of Neurobiology on Physical exercise and how it can be helpful in prevention of PTSD and CPTSD and how it can influence treatment of affected patients

6. Conclusion

• The key conclusion that can be drawn from the findings of the study are not well elaborated

• Grammar and tenses need to be addressed – See 2nd sentence

7. Acknowledgments: Missing

6. PLOS authors have the option to publish the peer review history of their article (what does this mean?). If published, this will include your full peer review and any attached files.

**Do you want your identity to be public for this peer review?** For information about this choice, including consent withdrawal, please see our Privacy Policy.

Reviewer #1: No

Reviewer #2: No

Reviewer #3: No

---

## [Decision Letter · Decision Letter 1]

9 Jul 2024

PMEN-D-24-00076R1

Trauma clinicians' views of physical exercise as part of PTSD and Complex PTSD treatment: A qualitative study

PLOS Mental Health

Dear Dr. Billings,

Thank you for submitting your manuscript to PLOS Mental Health.  The reviewers believe that all major issues have been resolved. However, there are some minor issues that still need to be addressed, as outlined below. Therefore, we invite you to submit a revised version of the manuscript that addresses the points raised during the review process.

We look forward to receiving your revised manuscript.

Kind regards,

Hongru Zhu

Academic Editor

PLOS Mental Health

Journal Requirements:

https://journals.plos.org/mentalhealth/s/figures 

https://journals.plos.org/mentalhealth/s/figures#loc-file-requirements 

Additional Editor Comments (if provided):

Reviewers' comments:

Reviewer's Responses to Questions

**Comments to the Author**

1. If the authors have adequately addressed your comments raised in a previous round of review and you feel that this manuscript is now acceptable for publication, you may indicate that here to bypass the “Comments to the Author” section, enter your conflict of interest statement in the “Confidential to Editor” section, and submit your "Accept" recommendation.

Reviewer #3: All comments have been addressed

2. Does this manuscript meet PLOS Mental Health’s publication criteria? Is the manuscript technically sound, and do the data support the conclusions? The manuscript must describe methodologically and ethically rigorous research with conclusions that are appropriately drawn based on the data presented.

Reviewer #3: Yes

3. Has the statistical analysis been performed appropriately and rigorously?

Reviewer #3: Yes

4. Have the authors made all data underlying the findings in their manuscript fully available (please refer to the Data Availability Statement at the start of the manuscript PDF file)?

Reviewer #3: Yes

5. Is the manuscript presented in an intelligible fashion and written in standard English?

Reviewer #3: Yes

6. Review Comments to the Author

Reviewer #3: Comments:

Thanks for the revision and most of the comments have been addressed by the authors. However, some minor one below need to be addressed

1. General comments and some key concerns:

• Use past tense instead of future tense in sentence construction in the document

• Limit use of first and second person in sentence construction. Instead use third person.

2. Methods section

• Insert “Methodology” instead of “Methods”

• Which statistical package was used to summarize the data in table 1? Is this data qualitative or quantitative?

4. Results

• Responses in table 1- Insert “Responses (%, n)” as sub-title in the cell and then remove % on each of the figures

• The % should come first i.e. 75.0 (9) and Write % to 1 decimal place all through the table

7. PLOS authors have the option to publish the peer review history of their article (what does this mean?). If published, this will include your full peer review and any attached files.

**Do you want your identity to be public for this peer review?** For information about this choice, including consent withdrawal, please see our Privacy Policy.

Reviewer #3: No

---

## [Editor Report · Decision Letter 2]

31 Jul 2024

Trauma clinicians' views of physical exercise as part of PTSD and Complex PTSD treatment: A qualitative study

PMEN-D-24-00076R2

Dear Dr Billings,

We are pleased to inform you that your manuscript 'Trauma clinicians' views of physical exercise as part of PTSD and Complex PTSD treatment: A qualitative study' has been provisionally accepted for publication in PLOS Mental Health.

Best regards,

Hongru Zhu

Academic Editor

PLOS Mental Health